# Sepsis risk in diabetic patients with urinary tract infection

**Sipei Wang**[1], **Sheng Zhao**[1], **Shanshan Jin**[1], **Tinghua Ye**[1], **Pan Xinling**[2]*

1 Department of Clinical Laboratory, Wenzhou Medical University Affiliated Dongyang Hospital, Dongyang, Zhejiang, China, 2 Department of Biomedical Sciences Laboratory, Wenzhou Medical University Affiliated Dongyang Hospital, Dongyang, Zhejiang, China

☯ These authors contributed equally to this work.
* panfengyuwuzu@163.com

## Abstract

### Background

Urinary tract infections (UTI) is a prevalent condition in those with diabetes, and in severe cases, it may escalate to sepsis. Therefore, it is important to analyze the risk variables associated with sepsis in diabetes individuals with UTI.

### Methods

This research was a retrospective cross-sectional analysis. From January 2011 to June 2022, a group of individuals with diabetes were identified as having UTI at a tertiary hospital situated in Southeastern China. Patient data, including information on urine culture, was collected retrospectively from a clinical record database. The participants were categorized into the sepsis and non-sepsis groups. The risk variables were derived using both uni-and multiple- variable regression analysis.

### Results

The research included 1919 patients, of whom 1106 cases (57.63%) had positive urine cultures. In total, 445 blood culture samples were tested, identifying 186 positive cases (41.80%). The prevalence of bacteria in urine and blood samples was highest for *Escherichia coli* and *Klebsiella pneumoniae*, respectively. Moreover, 268 individuals (13.97%) exhibited sepsis. The regression analysis indicated a positive correlation between sepsis and albumin (ALB)<34.35 g/L, C-reactive protein (CRP)>55.84 mg/L and white blood cell count (WBC) >8.485 X $10^9$/L in diabetic cases with UTIs. By integrating the three aforementioned parameters, the area under the receiver operating characteristic curve was 0.809.

### Conclusions

The early detection of sepsis in diabetic individuals with UTI may be achieved using a comprehensive analysis of CRP, WBC, and ALB test findings.

**Data Availability Statement:** All relevant data are within the manuscript and its Supporting Information files.

**Funding:** This study was supported by the Fundings from the Science and Technology Bureau of Jinhua (grant No. 2022-4-278 and 2023-3-031).

The funders had a role in the data collection and analysis.

**Competing interests:** We declare that no conflict of interest exits in the submission of this manuscript, and manuscript is approved by all authors for publication.

## Introduction

Diabetes is a collection of metabolic disorders distinguished by the persistent elevation of blood glucose levels. Over the last three decades, there has been a substantial rise in the incidence of diabetes among the Chinese population. According to a study by the Endocrinology Branch of the Chinese Medical Association between 2015 and 2017, the prevalence of diabetes among adults aged 18 and older in China was 11.2% [1].

Diabetes might be complicated by various illnesses, leading to a vicious cycle of infections and uncontrolled hyperglycemia [2]. UTI is prevalent in this population, constituting a significant proportion of infections. Infection can trigger acute problems in individuals with diabetes, hence contributing significantly to mortality rates. In comparison to those without diabetes, patients with diabetes had a 1.21–2.2 times higher relative risk of UTI [3]. Community-acquired UTI is a primary contributor to bloodstream infections within the community, constituting around 30–35% of bacteremia cases among adults [4]. Among patients with sepsis, it has been shown that the most common infection site is urinary system, following the respiratory tract and abdomen [5]. In the case of patients with UTI, diabetes has been identified as a distinct risk factor for the development of urosepsis [6]. This association may be attributed to the impaired immune function often seen in individuals with diabetes [7].

Sepsis is a critical and potentially fatal consequence that necessitates prompt identification of affected individuals to facilitate the implementation of early and efficacious treatment strategies. A delay of one hour in the administration of antibiotics for urinary-derived sepsis was reportedly associated with a reduction in patient survival rate by 7.6% [8].Therefore, the incidence of sepsis may be reduced by implementing timely risk assessment and appropriate antibiotic therapy. There are literature reports on the identification of risk factors for progression to sepsis in patients with different diseases [9,10], but there are few reports on progression to sepsis in patients with diabetes mellitus complicated with UTI. Hence, our objective was to ascertain certain indications that might be used to promptly identify the distinguishing features of diabetic individuals with UTI exacerbated by sepsis. This endeavor aims to provide valuable insights for clinical diagnosis and therapy.

## Materials and methods

### Patients' inclusion and exclusion criteria

From January 2011 to June 2022, the data of individuals with diabetes and UTI, that were admitted to Dongyang People's Hospital, were retrospectively collected from the clinical record database if they satisfied the diagnostic criteria. Individuals were diagnosed with diabetes upon meeting one of the aforementioned criteria [11]: (1) fasting blood glucose level $\geq 7.0$ mmol/L, (2) 2-hour postprandial blood glucose level $\geq 11.1$ mmol/L, (3) random blood glucose level $\geq 11.1$ mmol/L, (4) glycated haemoglobin (HbA1c) $\geq 6.5\%$. The confirmation of a UTI diagnosis may be achieved if patients exhibit one of the following conditions: (1) typical symptoms of UTI + pyuria (defined as >5 white blood cells WBCs/HP) + positive urine nitrite test, (2) WBCs >10/HP in the urine sediment obtained from the middle part of a cleaning centrifuge/typical symptoms of UTI + urinary bacterial count $\geq 10^5$/ml, (3) urinary bacterial count $\geq 10^5$/ml for two consecutive times, and the bacteria and subtypes of the two times were the same, (4) the urine obtained from a bladder puncture showed positive results for culture, (5) typical symptoms of UTI + early morning clean midstream urine centrifuge urine sediment Gram stain before treatment to find bacteria (bacteria >1/oil mirror field)[12]. The exclusion criteria for this study included the existence of an immunodeficiency condition.

## Specimen collection, bacterial culture and species identification

The process of collecting and transferring specimens adhered to the established guidelines of the health industry in the People's Republic of China, namely the WS/T640-2018 standard for specimen collection and transit in clinical microbiology. Briefly, urine samples of 20–50 mL were collected using sterile containers, while blood samples of 5–10 mL were taken in a blood culture vial manufactured by bioMérieux, France. The specimens were maintained under ambient conditions and promptly transported to the laboratory within 2 hours. The specimens were subjected to culturing on blood agar and Mackand agar plates (Kangtai, Wenzhou). Subsequently, they were incubated at 35˚C under a 5% $CO_2$ atmosphere for 24 to 48 hours. The species identification was conducted via a Vitek2 Compact system and matrix-assisted laser desorption/ionization time-of-flight mass spectrometry (MALDI-TOF MS) when visible colonies were seen on the medium plates. The strains used for quality control in this study included *Klebsiella oxytoca* ATCC700324, *Staphylococcus. saprophyticus* ATCCBAA750, and *Escherichia coli* ATCC8739.

## Data collection

The retrospective acquisition of patients' information was conducted from a clinical record database, encompassing age, gender, medical history of hypertension, coronary heart disease, malignant tumor, complications related to diabetes, utilization of glucocorticoids and catheterization, procalcitonin (PCT) levels upon initial admission, c-reactive protein (CRP), ALB, WBC, HbA1c, urine leukocyte (LEU), urine glucose levels, presence of sepsis, and blood culture results. The diagnostic criteria for sepsis were used in accordance with the guidelines outlined in the Third International Consensus Definitions for Sepsis and Septic Shock [13].

## Statistical analysis

The statistical analysis was conducted via SPSS 26.0. The counting data was presented in cases and percentages. The analysis included the use of the Chi-square test to investigate the differences between the two groups. In cases of a normal distribution, The continual data with normal distribution was often denoted as mean ± standard deviation, whereas the data with non-normal distribution was represented by medians accompanied by quartiles. The difference on continual data between groups was examined via an independent sample t-test (for data exhibiting a normal distribution) or a u-test (for data displaying a non-normal distribution). Continuous variables with P values less than 0.05 were then included in a univariable logistic regression analysis, and the cut-off values were established using the maximum Youden index. The significant variables identified in the previous analysis were included in a multivariable logistic regression model, using a forward direction approach. These variables were then merged to assess their potential utility in distinguishing between the two groups, as measured by the area under the receiver operating characteristic curve.

## Ethics approval

This study involved human participants and was approved by the Ethics Committee and Institutional Review Board of Dongyang People's Hospital (No. 2023-YX-409). The data were collected and analyzed anonymously. The need for informed consent was waived due to the retrospective nature of the study and all methods were carried out in accordance with relevant guidelines and regulations.

## Results

### Urine and blood culture in diabetes patients with UTI

All 1919 patients were tested for urine culture, including 1106 (57.63%) positive cases. Moreover, blood culture was tested in 445 cases, with 186 (41.80%) positive cases.

### Isolation of pathogenic bacteria in urine and blood

A comprehensive collection of 1185 strains of pathogenic bacteria were obtained from 1106 urine samples, while 186 bacteria isolates were identified from 186 blood culture samples. *Escherichia coli* was the most prevalent pathogenic bacterium found in urine and blood samples, with *Klebsiella pneumoniae* being the subsequent most often seen. The distribution of strain species exhibited subsequent variations (**Fig 1**).

### Sepsis-related factors by univariable analysis

Among the 1919 diabetic cases with UTI, 268 developed sepsis as a complication. The results of the univariate analysis indicated that the sepsis group exhibited lower ALB concentration compared to the non-sepsis group. Conversely, the sepsis group had greater levels of CRP, PCT, HbA1c, and WBC in comparison to the non-sepsis group. The prevalences of patients in the sepsis group exhibiting positive LEU, positive urine glucose, catheter usage, and glucocorticoid use were greater (**Table 1**).

### The cut-off value of enrolled continual variables in the univariate analysis

The recorded cut-off values for ALB, CRP, PCT, HbA1c, and WBC were 34.35 g/L, 55.84 mg/L, 1.015 ng/ml, 8.35% and $8.485X10^9$/L, respectively. The values corresponding to the AUCs were 0.782, 0.787, 0.849, 0.595, and 0.731, respectively (**S1 Table**).

### The independent risk factors for sepsis by multivariate logistic regression analysis

PCT was excluded from further investigation because of a missing rate of 47.68%. Subsequently, a multivariate logistic regression analysis was conducted, including ALB, CRP, HbA1c, WBC, LEU, urine glucose, coronary heart disease, catheterization, and glucocorticoid usage as variables. The findings indicated that diabetic patients with UTI who exhibited ALB<34.35 g/L, CRP>55.84 mg/L, and WBC>$8.485X10^9$/L were identified as risk factors for sepsis (**Table 2**).

Based on the aforementioned parameters, the AUC of logistic regression model was 0.809, with a standard error of 0.016. Additionally, the asymptotic 95% Confidence Interval ranged

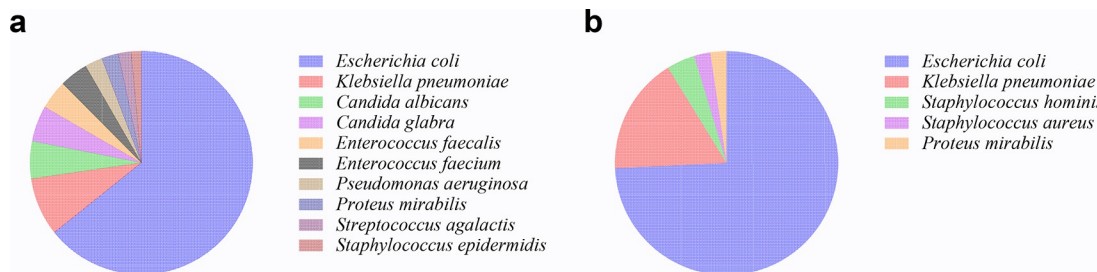

**Fig 1. Isolation of pathogenic bacteria in urine and blood samples.** (a) the top ten bacteria isolated from urine samples; (b) the top five bacteria isolated from blood samples.

**Table 1. Univariable analysis of the differences in sepsis and non-sepsis groups.**

| Variables | sepsis (n = 268) | Non-sepsis (n = 1651) | Z | χ2 | P |
|---|---|---|---|---|---|
| Age (year) | 70 (59, 78) | 70 (61, 79) | -1.331 | | 0.183 |
| Male in Gender, n (%) | 100 (37.31) | 596 (36.10) | | | 0.701 |
| ALB (g/L) | 31.4 (28.6, 34.3) | 37.1 (33.6, 40.2) | -12.209 | | <0.001 |
| CRP (mg/L) | 111.63 (43.33, 184.63) | 13.59 (2.5, 58.8) | -14.488 | | <0.001 |
| PCT (ng/ml) | 3.265 (0.771, 13.25) | 0.166 (0.061, 0.566) | -16.445 | | <0.001 |
| HbA1c (%) | 7.7 (6.5, 9.8) | 7 (6.2, 8.7) | -4.326 | | <0.001 |
| WBC ($10^9$/L) | 12.04 (8.76, 17.56) | 7.56 (5.87, 10.77) | -12.078 | | <0.001 |
| LEU positive in urine, n (%) | 211 (81.78) | 1171 (73.51) | | 8.035 | 0.005 |
| Glucose in urine n (%) | | | | 19.105 | 0.002 |
| 4+ | 61 (22.76) | 252 (15.52) | | | |
| 3+ | 29 (10.82) | 142 (8.74) | | | |
| 2+ | 12 (4.48) | 79 (4.86) | | | |
| 1+ | 27 (10.07) | 106 (6.53) | | | |
| ± | 6 (2.24) | 67 (4.13) | | | |
| - | 133 (49.63) | 978 (60.22) | | | |
| Complications of diabetes, n (%) | 48 (17.91) | 320 (19.38) | | 0.322 | 0.570 |
| Hypertension, n (%) | 160 (59.7) | 1084 (65.66) | | 3.587 | 0.058 |
| Malignant tumor, n (%) | 25 (9.33) | 194 (11.75) | | 1.338 | 0.247 |
| Coronary heart disease, n (%) | 39 (14.55) | 354 (21.44) | | 6.72 | 0.010 |
| Catheterization, n (%) | 23 (8.58) | 74 (4.48) | | 8.076 | 0.004 |
| Glucocorticoid use, n (%) | 70 (26.12) | 241 (14.6) | | 22.541 | <0.001 |

ALB, albumin; CRP, C-reactive protein; PCT, procalcitonin; HbA1c, glycated haemoglobin; WBC, white blood cell count; LEU, leukocyte; glucose in urine 4+, glucose concentration ≥2000 mg/dL; 3+, 500–1999 mg/dL; 2+, 250–499 mg/dL; 1+, 100–249 mg/dL; ±, 50–99 mg/dL; -, < 50 mg/dL.

from 0.778 to 0.839. Furthermore, the sensitivity of the model was 0.821, while the specificity was 0.673 (**Fig 2**).

## Discussion

In the present investigation, a total of 268 cases, accounting for 13.97% of patients with both diabetes and UTIs, were found to be concomitant with sepsis. The important risk factors associated with sepsis during hospitalization were the levels of ALB, CRP, and WBC.

The prevalence of pathogenic bacteria in urine was mostly attributed to *Escherichia coli*, a finding that aligns with the strains identified in urinary tract samples documented by CHINET in 2022 [14]. However, based on our collected data, *Klebsiella pneumoniae* and yeasts ranked second in terms of prevalence. In contrast, the CHINET data indicated that *Enterococcus* and *Klebsiella pneumoniae* were the predominant species. This may be attributed to the distinctive characteristics within our patient community, as individuals with diabetes have a higher

**Table 2. The risk factors for sepsis in diabetic patients with urinary tract infection by multivariable analysis.**

| Variables | β | SB | Walsχ2 | P | Exp(B) | Exp(B) 95% CI |
|---|---|---|---|---|---|---|
| ALB <34.35 g/L | 1.219 | 0.245 | 24.747 | 0.000 | 3.385 | 2.094–5.473 |
| CRP >55.84 mg/L | 1.229 | 0.256 | 22.977 | 0.000 | 3.418 | 2.068–5.650 |
| WBC >8.485 $10^9$/L | 1.087 | 0.269 | 16.299 | 0.000 | 2.965 | 1.749–5.026 |

ALB, albumin; CRP, C-reactive protein; WBC, white blood cell count.

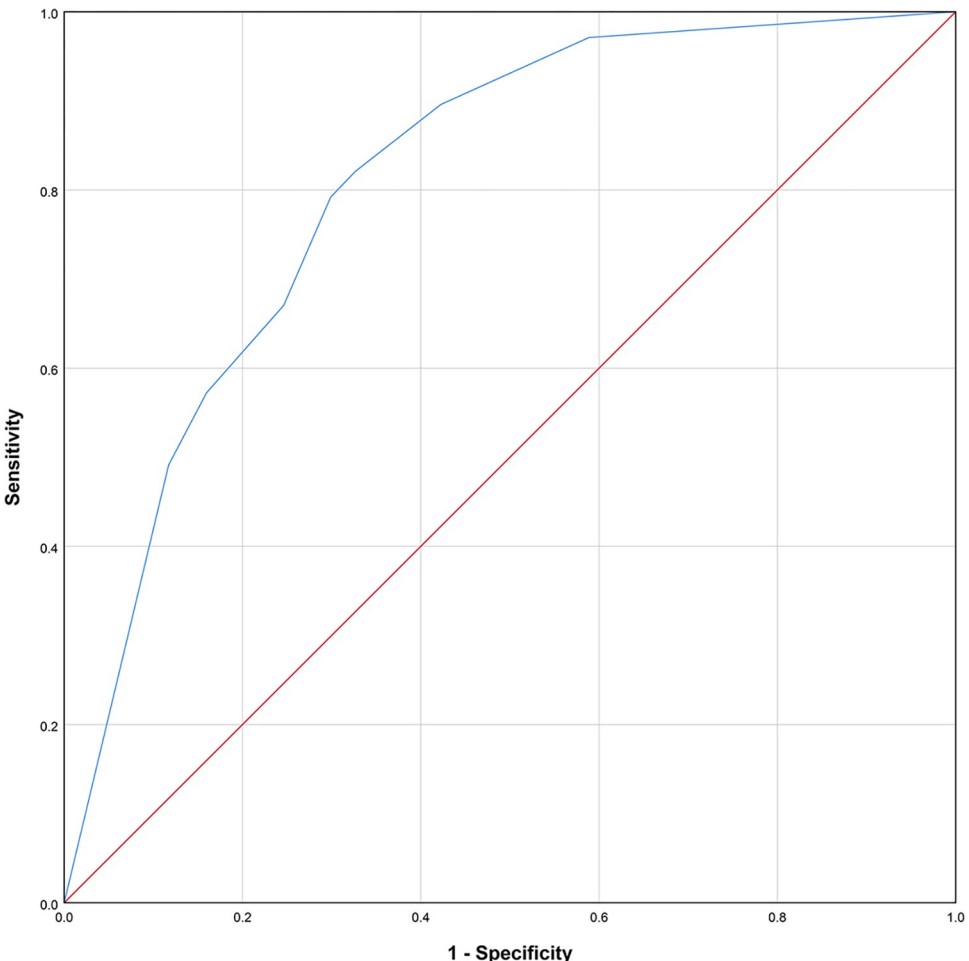

**Fig 2. ROC analysis of logistic regression based on the combination of CRP, WBC and ALB.**

susceptibility to opportunistic infections. It is noteworthy that yeast is among the primary pathogens associated with nosocomial opportunistic infections [15]. The microbial composition of pathogenic bacteria obtained from blood samples of individuals with diabetes exhibited distinct distributions compared to those present in urine samples. The first two pathogens identified were *Escherichia coli* and *Klebsiella pneumoniae*, subsequently succeeded by coagulase negative *Staphylococcus* and *Staphylococcus aureus*. The observed distribution of strains in the bloodstream of individuals with diabetes exhibited a high degree of conformity with the data provided by CHINET. The presence of coagulase negative *Staphylococcus* in a blood culture sample should be considered as a potential indication of contamination during a surgical procedure. *Staphylococcus aureus* has several virulence factors to enhance the strain's capacity to invade the bloodstream [12].

ALB is the predominant protein found in plasma, making it a valuable marker for assessing the nutritional status of individuals in clinical settings. The substance has antioxidant properties and also functions as a ligand after binding to bacteria [16]. Our study indicates that ALB is a significant contributing factor to the development of sepsis in individuals with diabetes and UTI. This relationship may be attributed to three possible mechanisms. First, albumin serves as a nutritional indicator, and individuals exhibiting hypoalbuminemia are indicative of suboptimal nutritional health. Second, albumin has distinct pharmacological characteristics

such as antioxidant capabilities and the ability to operate as a transporter. Consequently, a deficiency in albumin may potentially impact the efficacy of the agent-containing treatment. Third, albumin is recognized as a negative acute phase protein. Consequently, the presence of hypoalbuminemia in patients may indicate an elevated inflammatory condition, which might possibly result in unfavorable consequences [17].

This study found that diabetic individuals with UTIs and sepsis had substantially higher levels of CRP and WBC and a lower level of ALB compared to those without sepsis. However, the only usage of PCT had the AUC over 0.8, suggesting that it was a powerful indicator for recognizing sepsis compared to CRP. This finding is similar to previous research [18]. A prospective multicenter cohort research revealed that a PCT cut-off value of 0.25 ng/mL could effectively detect the presence of bacteremia in patients with febrile UTIs. The sensitivity o was 95%, while the specificity was 50% [19]. Our findings revealed a PCT cut-off value of 1.015, with a sensitivity of 0.717 and a specificity of 0.847. These results are consistent with the findings reported by Zhang et al, who set a PCT cut-off value of 1.51 [20]. Nevertheless, the testing results for ALB, WBC and CRP could be obtained within a shorter duration comparing to PCT, resulting into an earlier evaluation of sepsis risk in diabetic patients with UTI. Fortunately, the combination of ALB, WBC and CRP could predict the risk with an AUC of 0.809, comparable to only PCT. Hence, it is possible to use the parameters of WBC, CRP and ALB as indicators to estimate the likelihood of sepsis in patients, prior to the acquisition of PCT test results.

The formation of HBA1c via non-enzymatic reactions is a continuous, gradual, and irreversible process. As a result, the content of HBA1c is influenced by historical blood glucose levels rather than the instant concentration. It is independent of circumstances such as fasting, insulin injection, and the use of hypoglycemic medicines prior to measurement. The prevailing consensus is that the concentration of HbA1c is a reliable indicator of the mean blood glucose level over 8 to 12 weeks. This study indicates that the HbA1c level was a significant predictor associated with sepsis in diabetic patients with UTIs, after the exclusion of infection-related blood indicators (CRP and WBC) (S2 Table). However, the AUC was 0.782 (95% confidence interval: 0.739–0.824), indicating that it is less sensitive than the three-indicators model (S1 Fig)

The hyperglycemic condition seen in individuals with diabetes has been shown to have an impact on the functioning of immune cells. This, in turn, leads to a compromised immunological response, rendering diabetic patients unable to effectively defense the invasion of pathogens [21]. When blood glucose levels above the range of 160 to 180mg/dl, a significant amount of glucose is eliminated via the urine, resulting in the formation of glucosuria. The increased prevalence of individuals exhibiting positive urine glucose in the sepsis group is indicative of a heightened susceptibility to severe infection resulting from hyperglycemia. Moreover, the presence of an indwelling catheter facilitates bacterial colonization, leading to the formation of biofilm [22]. Additionally, it facilitates the retrograde flow of urine from the bladder to the kidneys, contributing to inhibiting the contraction of the ureter, hence exacerbating bacterial growth and elevating the susceptibility to UTI in conjunction with sepsis [23]. Based on our findings, urine glucose, cardiovascular disease, and catheterization do not exhibit a significant association with the risk of sepsis. However, it is worth noting that other researches have reported these characteristics as potential risk factors for sepsis [24,25]. Comparing to the blood markers, these indications may lack prediction power, hence resulting in their exclusion from the final model in this study.

However, this study has some limitations. The data was collected from single center and a possible reduced applicability of the findings might exist for other patient population. Due to a long study period in this retrospective study, the temporal bias on the diagnosis and treatment

for diabetes, sepsis and urinary tract infection could not be avoided. The modifications in the guidelines mainly have impact on the incidence of enrolled disease, but have a limited role in the finally enrolled risk factors related to sepsis. In addition, other risk factors (such as urinary system stones, blood creatinine, etc.) for the development of sepsis in patients with diabetes complicated with urinary tract infection were not included in the study due to excessive absence from the clinical record database.

## Conclusion

In a cohort of individuals with diabetes and UTIs, the predominant causative agent identified in urine samples and blood samples were *Escherichia coli* and *Klebsiella pneumoniae*. A total of 13.97% of the patients in the study had sepsis and the risk of sepsis was related to ALB<34.35 g/L, CRP>55.84 mg/L, WBC>8.485 X 10^9/L. The early detection of sepsis in diabetic individuals with urinary tract infection may be achieved by using a combination of CRP, WBC, and ALB test findings.

## Supporting information

**S1 Fig. The ROC curve of the prediction model based on a combination of ALB and HbA1c.**
(TIF)

**S1 Table. The cut-off values of variables related to sepsis.**
(DOCX)

**S2 Table. The risk factors related to sepsis excluding inflammatory indices (CRP, WBC).**
(DOCX)

**S1 Raw data.**
(XLS)

## Author Contributions

**Data curation:** Sheng Zhao, Shanshan Jin.

**Formal analysis:** Tinghua Ye.

**Writing – original draft:** Sipei Wang.

**Writing – review & editing:** Pan Xinling.

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
