## [Decision Letter · Decision Letter 0]

8 Feb 2024

PONE-D-23-44217Sepsis risk in diabetic patients with urinary tract infectionPLOS ONE

Dear Dr. Xinling,

Thank you for submitting your manuscript to PLOS ONE. After careful consideration, we feel that it has merit but does not fully meet PLOS ONE’s publication criteria as it currently stands. Therefore, we invite you to submit a revised version of the manuscript that addresses the points raised during the review process.

We look forward to receiving your revised manuscript.

Kind regards,

Seth Agyei Domfeh, PhD

Academic Editor

PLOS ONE

“This work was supported by Key Project of Science and Technology Bureau of Jinhua, (No. 2023-3-031) and the Science and Technology Bureau of Jinhua (No. 2022-4-278)”

“This study was supported by Key Project of the Science and Technology Bureau of Jinhua (grant No. 2023-3-031) and the Science and Technology Bureau of Jinhua (grant No. 2022-4-278).”

“This work was supported by Key Project of Science and Technology Bureau of Jinhua, (No. 2023-3-031) and the Science and Technology Bureau of Jinhua (No. 2022-4-278)”

5. We notice that your supplementary tables are included in the manuscript file. Please remove them and upload them with the file type 'Supporting Information'. Please ensure that each Supporting Information file has a legend listed in the manuscript after the references list.

6. We are unable to open your Supporting Information file [Raw Data.sav]. Please kindly revise as necessary and re-upload.

Reviewers' comments:

Reviewer's Responses to Questions

**Comments to the Author**

1. Is the manuscript technically sound, and do the data support the conclusions?

Reviewer #1: Yes

Reviewer #2: Yes

Reviewer #3: Yes

2. Has the statistical analysis been performed appropriately and rigorously? 

Reviewer #1: Yes

Reviewer #2: Yes

Reviewer #3: Yes

3. Have the authors made all data underlying the findings in their manuscript fully available?

Reviewer #1: Yes

Reviewer #2: Yes

Reviewer #3: Yes

4. Is the manuscript presented in an intelligible fashion and written in standard English?

Reviewer #1: Yes

Reviewer #2: Yes

Reviewer #3: Yes

5. Review Comments to the Author

Reviewer #1: General impression:

The research paper provides a concise overview of the study conducted on diabetic individuals with urinary tract infections (UTIs) and their risk of developing sepsis. The article effectively communicates the research objectives, methodology, key findings, and conclusions of the study in a clear and concise manner.

The investigation was carried out to a very high technical standard, as evidenced by the article's thorough explanations of the procedures employed for specimen processing, data collecting, and statistical analysis. Making these points clear improves the findings' repeatability and dependability.

The statistical techniques applied, such as the usage of SPSS software and different tests including logistic regression, t-test, and Chi-square, are suitable for the analysis of the data. The explanation of the treatment of continuous variables according to their distribution clarifies the analysis procedure.

As is customary with categorical data, the article discusses the display of counting data in cases and percentages. When dealing with non-normally distributed data, the usage of medians and quartiles is appropriate.

It is laudable that a multivariate logistic regression model with a forward selection strategy was included in order to find important factors related to the desired outcomes. A reliable technique for assessing prediction models is the application of ROC curve analysis to determine the discriminatory strength of the selected variables.

The statistical analysis performed for the study lends support to the appropriateness of the conclusions derived from the data.

Minor issues

1. It is imperative to state the study's limitations, including the possibility of biases present in retrospective studies and the applicability of the findings to larger patient populations.

2. The study runs from January 2011 to June 2022, or more than ten years. If the analysis does not sufficiently address or account for changes in clinical practices, diagnostic criteria, or treatment procedures over time, temporal bias may be introduced. Authors should kindly state whether any notable modifications to the treatment of sepsis, UTIs, or diabetes occurred throughout the study period that would have affected the results found.

Reviewer #2: The article written by Wang et al. analysed the evolution of sepsis for diabetic patients with urinary tract infections. The title and the abstract are appropriate for the content of the text.Furthermore, the article is well constructed and analysis was well performed.Even if the statistics is very simple, the results obtained are very clear and demonstrate a relation between sepsis and biochemical parameters. The article highlights important data related to CRP, WBC and ALB which can be used to detect sepsis from early stages.

Reviewer #3: Peer Review Report

Manuscript #: PONE-D-23-44217

Peer Review Report on “Sepsis risk in diabetic patients with urinary tract infection”

1. Original submission

1.1 Recommendation

Minor revision

2. Comments to Authors

This study was performed to analyze the risk variables associated with sepsis in diabetes individuals with urinary tract infection (UTI) in China. The manuscript is fairly written but the authors need to make some minor revision to the manuscript in order to improve the manuscript significantly.

Below are details of my concerns and I ask that the authors specifically address each of my concern or comments in their response.

Line 65: change is after objective to was

Line 72: change was before retrospectively to were

Line 77: replace the full stop after nmol/L with comma

Line 81: replace the full stop after ≥105 mL with comma

Line 83: replace full stop after same with comma

Line 84: replace full stop after culture with comma

Lines 84 to 86: consider revising the sentence beginning typical symptoms of UTI…….. as it not clear. Also early morning cleaning in the sentence should read early morning clean, gram should be written as Gram

Line 97: write CO2 as CO2

Line 101: change include to included; you either write Stap in full or use S. saprophyticus

Line 128: change involves to involved

Provide appropriate heading for Table 1, for example Table 1…………………….the table heading should be placed on top of the Table

Line 165: write table with initial upper-case letter

Line 175: The Table 2 heading is not appropriate so change it. It could be written as Table 2: Risk factors for sepsis in diabetic patients

Line 191: write enterococcus with initial upper-case letter

199: italicize Staphylococcus aureus

Line 221: change with to ‘to’

Line 229: change leads to lead

Line 246: change Supplementary table 2 to Supplementary Table 2

Line 248: change Supplementary fig 1 to Supplementary Fig 1

Line 249: change table to Table

Properly label the Table on page 11 with proper heading

Line 252: change fig to Fig

Page 23: italicize all the scientific names of bacteria besides the pie chart

---

## [Author Response · Author response to Decision Letter 0]

25 Apr 2024

Reviewer #1: General impression:

The research paper provides a concise overview of the study conducted on diabetic individuals with urinary tract infections (UTIs) and their risk of developing sepsis. The article effectively communicates the research objectives, methodology, key findings, and conclusions of the study in a clear and concise manner.

The investigation was carried out to a very high technical standard, as evidenced by the article's thorough explanations of the procedures employed for specimen processing, data collecting, and statistical analysis. Making these points clear improves the findings' repeatability and dependability.

The statistical techniques applied, such as the usage of SPSS software and different tests including logistic regression, t-test, and Chi-square, are suitable for the analysis of the data. The explanation of the treatment of continuous variables according to their distribution clarifies the analysis procedure.

As is customary with categorical data, the article discusses the display of counting data in cases and percentages. When dealing with non-normally distributed data, the usage of medians and quartiles is appropriate.

It is laudable that a multivariate logistic regression model with a forward selection strategy was included in order to find important factors related to the desired outcomes. A reliable technique for assessing prediction models is the application of ROC curve analysis to determine the discriminatory strength of the selected variables.

The statistical analysis performed for the study lends support to the appropriateness of the conclusions derived from the data.

Authors’ response: Thanks for positive comment from the reviewer. 

Minor issues

1. It is imperative to state the study's limitations, including the possibility of biases present in retrospective studies and the applicability of the findings to larger patient populations.

Authors’ response: We agreed to the reviewer’s comment that there were some limitations in this study, including the bias in retrospective studies and possible reduced applicability of the findings in other patient population. Thus, the limitations of this study were described in the revised manuscript. 

2. The study runs from January 2011 to June 2022, or more than ten years. If the analysis does not sufficiently address or account for changes in clinical practices, diagnostic criteria, or treatment procedures over time, temporal bias may be introduced. Authors should kindly state whether any notable modifications to the treatment of sepsis, UTIs, or diabetes occurred throughout the study period that would have affected the results found.

Authors’ response: Thanks for constructive comments. As the reviewer mentioned above, the clinical guidelines for diagnosis and treatment were reviewed to analyze the bias in this study. The diagnosis for sepsis was modified based on the sepsis 3.0, which was proposed in 2016. Thus, the data collected from 2011 to 2016 owned a significantly lower incidence of sepsis than that from 2017 to 2022. Another reason for the increasing incidence of sepsis was the stricter criteria for hospitalization, resulting into the fact that the patients who were accompanied by mild symptoms or less likely developed into worse status could not be hospitalized. However, the same analysis procedure was adopted in these two data (the groups admitted before and after 2016, respectively) and found that WBC, ALB and CRP were the consistent risk factors for sepsis. Therefore, the existence of bias from modification of sepsis diagnosis criteria did not change the risk factors enrolled in final prediction model. For the guidelines for UTI, there is nearly no modification. The diagnosis criteria for diabetes was modified in 2020, and adopted in clinical practice in 2021. The data from 2021 to 2022 accounted for 8% of total cases, limiting its influence on the findings. In conclusion, the bias in guidelines may have impact on the incidence of diseases, but had limited role in the risk factors related to sepsis risk in diabetic patients with UTI. The corresponding description was added in the limitation section. 

Reviewer #2: The article written by Wang et al. analysed the evolution of sepsis for diabetic patients with urinary tract infections. The title and the abstract are appropriate for the content of the text. Furthermore, the article is well constructed and analysis was well performed. Even if the statistics is very simple, the results obtained are very clear and demonstrate a relation between sepsis and biochemical parameters. The article highlights important data related to CRP, WBC and ALB which can be used to detect sepsis from early stages.

Authors’ response: Thanks for kind comments from the reviewer, and we believe our findings could be helpful for clinical staffs to evaluate the risk of sepsis in diabetic patients with urinary infection. 

Reviewer #3: Peer Review Report

Manuscript #: PONE-D-23-44217

Peer Review Report on “Sepsis risk in diabetic patients with urinary tract infection”

1. Original submission

1.1 Recommendation

Minor revision

2. Comments to Authors

This study was performed to analyze the risk variables associated with sepsis in diabetes individuals with urinary tract infection (UTI) in China. The manuscript is fairly written but the authors need to make some minor revision to the manuscript in order to improve the manuscript significantly.

Below are details of my concerns and I ask that the authors specifically address each of my concern or comments in their response.

Line 65: change is after objective to was

Line 72: change was before retrospectively to were

Line 77: replace the full stop after nmol/L with comma

Line 81: replace the full stop after ≥105 mL with comma

Line 83: replace full stop after same with comma

Line 84: replace full stop after culture with comma

Lines 84 to 86: consider revising the sentence beginning typical symptoms of UTI…….. as it not clear. Also early morning cleaning in the sentence should read early morning clean, gram should be written as Gram

Line 97: write CO2 as CO2

Line 101: change include to included; you either write Stap in full or use S. saprophyticus

Line 128: change involves to involved

Provide appropriate heading for Table 1, for example Table 1…………………….the table heading should be placed on top of the Table

Line 165: write table with initial upper-case letter

Line 175: The Table 2 heading is not appropriate so change it. It could be written as Table 2: Risk factors for sepsis in diabetic patients

Line 191: write enterococcus with initial upper-case letter

199: italicize Staphylococcus aureus

Line 221: change with to ‘to’

Line 229: change leads to lead

Line 246: change Supplementary table 2 to Supplementary Table 2

Line 248: change Supplementary fig 1 to Supplementary Fig 1

Line 249: change table to Table

Properly label the Table on page 11 with proper heading

Line 252: change fig to Fig

Page 23: italicize all the scientific names of bacteria besides the pie chart

Authors’ response: Thanks for detailed comments on our manuscript. We have checked the comments and modified them in revised text point by point. It was kind of the reviewer giving us helpful comments. Thanks again.

---

## [Editor Report · Decision Letter 1]

29 Apr 2024

Sepsis risk in diabetic patients with urinary tract infection

PONE-D-23-44217R1

Dear Dr. Xinling,

We’re pleased to inform you that your manuscript has been judged scientifically suitable for publication and will be formally accepted for publication once it meets all outstanding technical requirements.

Kind regards,

Seth Agyei Domfeh, PhD

Academic Editor

PLOS ONE

---

## [Editor Report · Acceptance letter]

4 May 2024

PONE-D-23-44217R1 

PLOS ONE

Dear Dr. Xinling, 

I'm pleased to inform you that your manuscript has been deemed suitable for publication in PLOS ONE. Congratulations! Your manuscript is now being handed over to our production team.

Kind regards, 

on behalf of

Dr. Seth Agyei Domfeh 

Academic Editor

PLOS ONE